# Environmental management accounting affects bank performance with mediators

**Kim Quoc Trung Nguyen** *

University of Finance – Marketing, Vietnam

* nkq.trung@ufm.edu.vn

## Abstract

This study aims to estimate the effect of environmental management accounting (EMA) on Vietnamese bank performance under the mediating role of environmental costs. The research employs qualitative methods, such as expert interviews and surveys, alongside quantitative methods, such as Partial least squares Structural Equation Modeling (PLS-SEM). The findings revealed a positive correlation between EMA and knowledge management, green innovation, and environmental costs. Additionally, knowledge management and green innovation significantly positively influence environmental costs. Significantly, the study emphasizes the relationship between EMA and a bank's performance, mediated by environmental costs. Recognizing the significance of environmental costs in the total cost structure, this study highlights their potential emergence in the provision of financial services. This study underscores the role of environmental accounting, which integrates the financial and management accounting aspects, in providing information on these costs.

## Introduction

Environmental pollution resulting from industrial activities is becoming increasingly severe. Methods for mitigating environmental pollution in business operations are of great concern to both enterprises and regulatory authorities. To address this issue, businesses need to understand the regulations, invest in upgrading waste treatment systems, and simultaneously implement environmental management accounting (EMA). Sustainable development is a process that meets current needs without compromising the interests of future generations. As a crucial and leading component of sustainable economic development, businesses are responsible for assessing and accounting for environmental factors to protect the environment.

The environment has significant importance in the realm of sustainable development. Organizations are increasingly prioritizing sustainable accounting, aiming to integrate environmental concerns into traditional accounting frameworks such as management accounting and EMA [1].

**Data availability statement:** Primary data can be found at the following web-site: https://datadryad.org/stash/share/KZxtpik1snnVXe6JPSHSNTyA7kLcFVj3_vmk-GKRKj54.

**Funding:** This research is funded by University of Finance – Marketing.

**Competing interests:** The authors have declared that no competing interests exist.

Vietnam's rapid industrialization and economic growth have exacerbated environmental challenges, compelling policymakers and businesses to prioritize sustainable practices. As the country strives to meet international sustainability commitments, the role of EMA becomes crucial for its ability to align financial strategies with environmental goals. However, despite its global recognition as a tool for improving both economic and environmental efficiency, the adoption of EMA in Vietnam remains limited. The lack of comprehensive frameworks, awareness, and regulatory incentives has hindered its implementation, particularly in the banking sector [2,3]. Many nations prioritize environmental preservation and fostering green growth as key objectives in their pursuit of sustainability. To ensure a balance between economic development and environmental protection, numerous studies worldwide have affirmed that EMA is a valuable tool for gaining a better understanding and quantifying environmental issues in decision-making processes [4]. EMA provides essential information to businesses to minimize environmental impacts, improve both economic and environmental efficiency, and achieve sustainability [5].

While global research has advanced EMA application, significant gaps exist in its integration within Vietnam's commercial banking sector. Unlike industrial enterprises, banks play a critical role in directing financial resources towards environmentally sustainable projects, making their adoption of EMA pivotal. However, existing studies have largely overlooked the sector-specific dynamics of Vietnamese banks, such as their potential to influence environmental outcomes through credit allocation and operational changes. For example, banks in Vietnam are in a unique position to fund environmentally sustainable projects, yet the lack of comprehensive EMA frameworks results in missed opportunities for aligning financial performance with environmental goals [6,7]. Additionally, there is limited research exploring how mediators like environmental costs, knowledge management, and green innovation amplify EMA's impact on performance, creating a critical knowledge gap that this study aims to address.

Furthermore, the Vietnamese banking sector is increasingly pressured to align with international sustainability standards. Global benchmarks, such as those set by the United Nations Environment Programme Finance Initiative (UNEP FI), require banks to adopt sustainable practices, yet Vietnamese banks lag behind in implementing EMA to meet these expectations. This delay not only affects their global competitiveness but also limits their ability to achieve long-term financial stability while addressing environmental challenges [3,8].

In addition, Vietnamese banks face distinct operational challenges that complicate EMA implementation. These include limited managerial expertise in environmental accounting, lack of technological support for tracking environmental costs, and inadequate stakeholder pressure compared to their counterparts in developed economies. Addressing these operational hurdles is critical to ensuring that banks not only meet international benchmarks but also leverage EMA to drive innovation and competitive advantage in a rapidly globalizing market [2,7]. Moreover, this study is particularly novel as it integrates sector-specific mediators—environmental costs, knowledge

management, and green innovation—into the analysis of EMA's impact on bank performance. These mediators have been underexplored in the Vietnamese context, despite their proven importance in global banking practices. By investigating their role, the study provides actionable insights for policymakers and banking professionals seeking to enhance both financial and environmental performance [6,8].

This study bridges these gaps by focusing on the Vietnamese banking sector, offering empirical insights into how EMA affects bank performance through mediators such as environmental costs, knowledge management, and green innovation. Unlike prior studies that have predominantly explored manufacturing contexts, this research highlights the strategic potential of EMA in enabling banks to meet sustainability goals while enhancing bank performance. The research is particularly relevant as Vietnamese banks face increasing pressure to comply with global environmental standards and adopt greener operational models [2,3].

Therefore, this study aims to estimate the effect of EMA on bank performance in the context of the mediating role of environmental costs. To achieve this objective, the following questions need to be addressed: What are the specific mechanisms through which EMA influences bank performance in Vietnam, and how do environmental costs, knowledge management, and green innovation act as mediators?

## Literature review and hypothesis development

### Literature review

**Legitimacy theory.** Legitimacy theory provides a framework for understanding how organizations justify their operations in alignment with societal expectations. It emphasizes the necessity for businesses to disclose information on social, economic, and environmental aspects to maintain their legitimacy [9]. EMA practices emerge as a strategic tool under this theory, helping organizations enhance their reputation, fulfill regulatory requirements, and secure societal approval [10]. By disclosing environmental information through EMA, firms demonstrate accountability and commitment to social norms, addressing critical expectations of regulators, investors, and customers. These efforts not only safeguard legitimacy but also position organizations as socially responsible entities. This is particularly critical for Vietnamese banks, which face increasing scrutiny from both regulators and international stakeholders [3].

**Contingency theory.** Contingency theory underscores the need for organizations to adapt their management systems to fit their unique operational contexts. The appropriateness of EMA depends on factors such as industry type, organizational size, and strategic priorities [11,12]. For Vietnamese banks, the integration of EMA practices must address sector-specific challenges like limited technological infrastructure, evolving regulatory demands, and the drive for sustainable development [6,8]. Contingency theory justifies tailoring EMA frameworks to include environmental costs and green innovation, ensuring that management accounting practices align with organizational goals while addressing external pressures for sustainability. This perspective reinforces the importance of flexibility in EMA adoption to enhance both environmental and financial outcomes.

**Stakeholder theory.** Stakeholder theory emphasizes the impact of diverse stakeholder groups on organizational decision-making processes and adherence to environmental laws [13,14]. This theory highlights the critical role of stakeholder engagement in promoting transparency and accountability through environmental cost management. Firms adopting EMA demonstrate their commitment to balancing stakeholder interests with environmental responsibilities, which can enhance trust, corporate reputation, and long-term viability. By integrating environmental accountability into their operations, organizations address stakeholder demands for responsible business practices and improved environmental outcomes. Vietnamese banks, in particular, are under increasing pressure from global initiatives like the United Nations Principles for Responsible Banking to align their operations with sustainability goals [15]. By integrating EMA into their practices, these banks can better meet stakeholder expectations while advancing their strategic objectives.

## Hypothesis development

**Environmental management accounting affects knowledge management.** In uncertain environments, organizations consider Knowledge Management (KM) a crucial asset for their continuity [16]. KM is pivotal in fostering innovation and leveraging internal and external knowledge within a firm [17] because it is instrumental in creating value and sustaining organizational growth in practical scenarios [18]. Specifically addressing environmental concerns, Environmental KM has proven effective in addressing environmental issues [19]. Critical KM practices such as knowledge acquisition, sharing, and application play a vital role in fostering sustainable corporate development [20]. Our study utilized environmental KM practices, including knowledge absorption, receptivity, and sharing, to assess EP.

KM involves actively managing, creating, sharing, coding, retaining, and acquiring knowledge within an organizational context [21]. Integrating a KM system into EMA functions is essential to ensure easy accessibility and value of accounting information. The contemporary competitive landscape has made manufacturing services more intricate and knowledge-intensive [21]. In this dynamic environment, knowledge assets have become increasingly vital for production organizations to meet their performance objectives [7]. Employing a KM strategy is necessary for developing environmentally friendly service plans. In recent years, there has been a noteworthy expansion in both the research and application of KM. Given this evolving scenario, understanding how manufacturing sectors respond to the intersections of EMA and KM has become a compelling area of interest [8].

The relationship between KM and EMA has been suggested by [8,22,23].

*Hypothesis H1: EMA positively affects KM in Vietnamese banking sector.*

## Environmental management accounting affects green innovation

To achieve environmental sustainability, organizations must incorporate green innovation into their existing operations, as highlighted by Li et al. [24]. Green innovation involves formulating and developing operations, including products, services, and processes, which result in less harm to the environment than available alternatives, as defined by Zeng et al. [25]. To implement green innovation effectively, firms must innovate in two primary areas: products and processes. This allows firms to minimize waste in their operations by maximizing resource utilization and reducing environmental pollution, aligning with organizational sustainability principles [26,27]. According to Ferreira et al. [28], green innovation can be categorized into three distinct types: green process, green product, and green management innovation.

In EMA implementation, research conducted by Saeidi et al. [29] indicate that EMA impacts both green products and process innovation. They concluded that a significant positive relationship exists between EMA utilization and innovation. Moreover, the empirical studies by Hadj; Huang and Li [30,31] also suggest that using the EMA is likely to yield positive outcomes for fostering green innovation.

*Hypothesis H2: EMA positively affects green innovation in Vietnamese banking sector.*

## Environmental management accounting affects environmental costs

EC is considered a component of accounting within the framework of sustainable development, aligned with the principles of environmental responsibility. They encompass the expenses businesses incur for managing their environmental impact, including costs associated with mitigation measures and compliance with environmental regulations. This definition places a corporate environmental protection obligation at its core to manage business activities to minimize the environmental impact while meeting environmental objectives. Environmental cost management aims to effectively manage business impacts on the environment and ensure compliance with environmental goals [32].

According to Schaltegger et al. [1], EC encompasses not only the direct expenses related to environmental protection but also other costs stemming from the EP of enterprises, enterprises' reputation, and enterprises' market value. In addition to environmental protection costs, these include a broader range of expenditure. Moreover, according to Elmaci

et al. [33], EC are defined as those that serve environmental protection goals. This implies that any expenditure on further environmental protection objectives can be considered as EC.

Businesses now recognize EC as expenses associated with the final waste disposal process, which are linked to environmental protection activities in compliance with environmental protection laws and some voluntary environmental expenses of the business [34].

Thus, EC has been identified as part of business expenses arising from the production process, which are the costs incurred by businesses to implement environmental protection measures. Identifying these costs helps businesses to comply with environmental laws, thereby achieving legal operations in business activities.

The EMA aims to integrate both financial and physical data, traditionally prioritizing the identification and minimization of environmental expenses, despite considering prospects and income. This emphasis on recognizing and lessening EC is evident in the evolution of EMA methodologies such as material flow cost accounting [2].

The majority of environmental expenses remain hidden and unrecognized, as they are categorized as overhead costs within conventional accounting frameworks [35,36].

Environmental concerns have mostly been ignored in the past when it came to industry and individual involvement [36,37]. Environmental activities seek to better use natural resources and reduce their impact on the environment [38]. Purchasing pollution control or reduction technologies, cleaning up the environment after ecological damage, protecting the economy from deteriorating environmental conditions, recycling, resource management, sustainability, and producing goods and services for the environment are a few examples [36]. According to [36,39], EMA tracks environmental efforts. Participation in environmental initiatives also has other benefits, such as improving client relations, product quality, and company reputation. Organizations are encouraged to make better decisions and perform better if hidden EC are disclosed. For example, the labor cost of maintaining equipment connected to the environment is usually not charged as an environmental cost [40].

According to Jasch [41], material efficiency, lower environmental effect and risk, and lower cost of environmental protection, EMA, or EMA, combines financial accounting, cost accounting, and material flow balance data. EMA is carried out by private or public companies, not by countries, and combines financial and physical elements. EMA measurements for internal decision-making encompass both monetarized metrics for costs, savings, and profits associated with actions that may influence the environment and physical metrics for material and energy consumption, flows, and final disposal. The evaluation of annual EC and expenditure is one of the main application areas for EMA data utilization.

As a component of EA, EMA is a useful tool for overcoming the drawbacks of traditional management accounting. It fosters greater appreciation for social responsibility and aids in the formulation of environmentally conscious business decisions [4]. EMA enhances a company's social standing and helps it manage expenses more effectively [42].

*Hypothesis H3: EMA positively affects environemtal costs in Vietnamese banking sector.*

## Knowledge management affects environmental costs

A KM system was designed to facilitate knowledge sharing and integration. This system acts as a repository for collecting, organizing, analyzing, and reusing knowledge that may be scattered throughout the organization. The main benefit of KM is that information is easily shared among the parties in an organization. Information in a company includes environmental information that can account for the EC [43]. Therefore, it is evident that businesses require information about EC to thoroughly evaluate the financial aspects of environmental management related to resource utilization and effectively manage the impact of knowledge. This, in turn, has a positive effect on EC [44].

Environmental cost accounting focuses on acknowledging and reclassifying environmental impacts and costs to facilitate improved decision-making processes [45]. The adoption of environmental cost accounting represents a proactive measure by companies to develop environmentally conscious accounting systems. It aids in gathering information pertinent to cost reclassification for environmental considerations, including material flows, social responsibility, and cost

accounting associated with sustainable development efforts [46]. When knowledge is well managed, the EC is fully disclosed, demonstrating the enterprise's responsibility to stakeholders. Thus, businesses need information about EC to fully evaluate the financial aspects of environmental management related to the use of resources, thus managing knowledge positively impact on EC.

According to Gurung & Landrum [47], environmental education within an organization, particularly in the knowledge-based economy era, can facilitate effective environmental KM, thereby enhancing the organization's intangible assets. Promoting environmental education within an organization fosters the professional development of environmental knowledge among its members and promotes positive environmental attitudes and professional performance, as highlighted by [48].

Furthermore, Chen & Tjosvold [49] suggest that employees' professional environmental performance can be improved by integrating environmental knowledge with environmental awareness, thus strengthening environmental beliefs and encouraging continuous environmental actions.

The contextual analysis revealed a positive correlation between KM and a facet of environmental performance. This indicates that implementing KM practices is crucial for enhancing EP across all the manufacturing processes. This improvement in EP can be quantified and assessed through measures such as cost, as highlighted by [50].

*Hypothesis H4: KM positively affects environemtal costs in Vietnamese banking sector.*

## Green innovation affects environmental costs

Financial and nonfinancial ratios are used to assess a company's success [51]. Additionally, companies may use green innovation to increase resource productivity and, in turn, compensate for their EC in terms of financial performance [52]. In addition to adhering to market competitiveness and demand dynamics, green innovation must address environmental protection concerns when inventing new goods, services, processes, and standards. As a result, focusing on green innovation raises firms' environmental expenses [53].

Previous research suggests that innovative green enterprises can mitigate regulatory costs while also generating additional profits [54,55]. Moreover, since resource wastage and energy loss are major contributors to pollution [56], adopting green process innovations can help businesses lower environmental costs by improving energy efficiency and promoting waste recycling [57,58].

Beyond these immediate cost reductions, achieving high levels of green process innovation offers significant long-term benefits, including lower environmental compliance costs [59], increased governmental support [60], and an enhanced green image for enterprises [61,62]. Financially, companies with a strong public image tend to be more highly valued by investors [63] and can attract new customers willing to pay premium prices for environmentally friendly products [64].

*Hypothesis H5: Green innovation positively affects environemtal costs in Vietnamese banking sector.*

## Environmental costs affect bank performance

Carroll [65] reveals a positive and significant relationship between high EP levels and overall corporate performance. Moreover, ENDIANA et al. [66] argued that discretionary investments in environmental improvement often yield financial benefits. Reducing pollution can lead to future cost savings by enhancing efficiency, decreasing environmental expenditure, and minimizing potential liabilities. Similarly, Elsheikhi et al. [67] asserted that companies adhering to stringent global environmental standards tend to have higher market valuations than those that do not.

Lee and Suh [68] claim a positive relationship between environmental control records and profitability. Dikgang et al. [69] highlight a more positive response from the stock market after environmental crises. Besides, Gehring et al. [70] upheld this argument, arguing that pollution control consumption and company profitability are not connected. Derila et al. [71] observe comparable outcomes and find that share returns, and EC have no direct relationship. Previous studies have recommended that the connection between the EC and financial performance should be clarified. Thus, stakeholder

theory shows companies as an influential aspect of a social system while concentrating on different stakeholder groups within the society [65].

Stakeholder theory examines the connection between EC and financial performance based on [72]. When these companies allocate resources to the EC, such as provisions or total liabilities, they demonstrate their commitment to environmental stewardship and financial sustainability. Managers focusing on operational excellence oversee the coordination of organizational processes, practices, policies, and relationships. According to this theory, companies are responsible for all stakeholders, including accountability for financial performance and fostering learning within the organization [72].

*Hypothesis H6: EC positively affects bank performance in Vietnamese banking sector.*

## Research methodology

**Research model.** Fig 1 shows the research model which determines the effect of the EMA on BP, under the mediating role of the KM, the GI, and the EC. Where the EMA is environmental management accounting; the KM is knowledge management; the GI is green innovation; the EC is environmental costs; and the BP stands for the bank performance.

## Methodology

The study employed a combined approach involving both qualitative and formal research methods. Data collection was conducted through verbal interviews with participants using qualitative techniques. Specifically, these individuals provided written informed consent, allowing the researchers to exclusively utilize the gathered information for this study and its subsequent publication in an academic journal. The interview participants typically included experts and professionals, such as head of accountants, managerial personnel from selected commercial banks in Ho Chi Minh City.

Participation in the study was secured by obtaining the interviewees' agreement to discuss banking-related information through a structured questionnaire during the in-depth interviews. This process aimed to identify the factors influencing the research and allow the information to be disclosed in this article. Furthermore, the interviewees consented to and assigned all rights to the authors to use the verbal data for this study. Additionally, all questions were designed based on the interviewees' knowledge and skills to provide evidence supporting the authors' assessments.

The University of Finance – Marketing granted ethical approval to the researchers. During data collection, the researchers adhered to the Decree on the Protection of Personal Data (Decree No. 13/2023/ND-CP, April 17, 2023), ensuring the anonymity and confidentiality of the respondents throughout the study.

This study comprised three stages, incorporating both qualitative and quantitative approaches. Initially, qualitative methods, including expert interviews with top managers, finance managers, chief accountants, and employee surveys, refined and enhanced observation factors on a preliminary scale. Based on the results of these interviews, group discussions form the primary scale. Subsequently, an interview questionnaire and surveys were developed to construct a

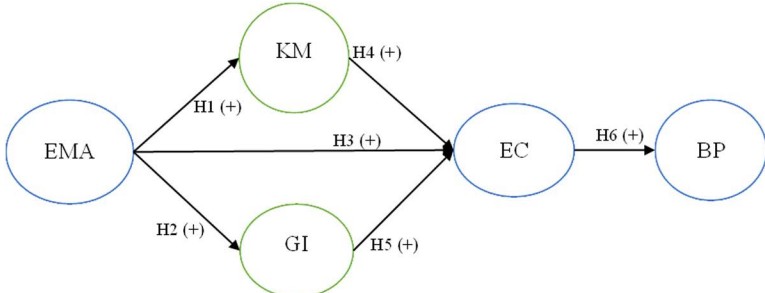

**Fig 1. Research model.**

formal questionnaire. During this phase, qualitative methods were utilized to identify pertinent factors within the current sector through interviews with ten experts from commercial banks. Following this, a group discussion involving 50 finance managers, chief accountants, and accountants was conducted to fine-tune the appropriate factor scales, ensuring that the survey questions were highly reliable [73]. All experts have agreed to engage in verbal discussion aimed at determining the primary measurement scale for analysis in the subsequent steps.

In the second stage, the author randomly surveyed respondents working at commercial banks in Vietnam using convenience sampling. In the last stage, the author conducts a quantitative research method using SmartPLS 4, which allows measurement and structural models to test the proposed hypothesis.

In addition, quantitative methods were applied to estimate the effect of the EMA on bank performance under EC intervention. The author collected, coded, and screened the data for analysis using SmartPLS 4. Partial least squares Structural Equation Modeling (PLS-SEM) was adopted to predict the research orientation and determine the findings.

### Research sample

According to Hair et al. [74], the minimum sample size in PLS-SEM determines using the "10-times rule" method. Hence, the small sample size (n=20 or less) was not valid for the PLS analysis. To satisfy the rule, 345 questionnaires were distributed to accountants at commercial banks, and the valid sample after filtering missing information was 312.

## Research results and discussions

### Research results

The first part of this section presents the demographic information of the participants based on sex and age, as shown in Table 1. Based on the statistical results of the respondent profile (312 respondents) in Table 2, 154 were male, accounting for 49.9%, and the remaining were female. Regarding age, the 36–45-year-old group had the highest proportion (62.9%), followed by the 46–55-year-old and 22–35-year-old group with 17% and 9.9%, respectively. The remaining patients were over 55 years of age.

To ensure the validity and reliability of the measurement model, the outer loadings for each indicator were assessed. Table 2 provides the factor loadings for the observed variables under their respective latent constructs (BP, EC, EMA, GI, and KM). According to Hair et al. [74], factor loadings ≥ 0.70 indicate that the indicators strongly contribute to the definition of their corresponding constructs.

The results in Table 2 show that all factor loadings meet or exceed the acceptable threshold, confirming the convergent validity of the constructs. Additionally, the high factor loadings reduce concerns about measurement error and enhance

**Table 1. Demographics of participants based on gender and age.**

| Particulars | Classes | Frequencies | % |
|---|---|---|---|
| Gender | | | |
| | Male | 154 | 49.4 |
| | Female | 158 | 50.6 |
| | Total | 312 | 100.0 |
| Age | | | |
| | 22–35 years | 31 | 9.9 |
| | 36–45 years | 216 | 69.2 |
| | 46–55 years | 53 | 17.0 |
| | Over 55 years | 12 | 3.8 |
| | Total | 312 | 100.0 |

the overall reliability of the model. These results support the appropriateness of the measurement model for further analysis.

The next section shows the construction reliability and validity, which are based on Cronbach's alpha, composite reliability (CR), and average variance extracted (AVE). Table 3 presents the results, providing an overview of the internal consistency, convergent validity, and the overall robustness of the measurement model.

The information provided in Table 3 shows composite reliability, such as Cronbach's alpha, composite reliability, and average variance extracted (AVE). According to Hair Jr et al. (2021), Cronbach's alpha for all factors is satisfactory. Furthermore, the composite reliability (CR) and average variance extracted (AVE) were greater than 0.7, and 0.5 respectively. Thus, it is evident that the establishment of reliability and convergent validity is satisfied [75].

Table 4 shows the discriminant validity using the Fornell-Larcker criterion. Fornell-Larcker compares the square root of the extracted AVE variance with the correlation coefficient of two latent variables [75]. According to Table 3, the indices on the diagonal of the Fornell-Larcker table are significantly greater than those below the diagonal, and the measurement instrument achieves discriminant validity [76].

**Table 2. Factor loadings.**

|  | BP | EC | EMA | GI | KM |
|---|---|---|---|---|---|
| BP1 | 0.923 |  |  |  |  |
| BP2 | 0.793 |  |  |  |  |
| BP3 | 0.915 |  |  |  |  |
| BP4 | 0.900 |  |  |  |  |
| EC1 |  | 0.962 |  |  |  |
| EC2 |  | 0.948 |  |  |  |
| EC3 |  | 0.952 |  |  |  |
| EC4 |  | 0.966 |  |  |  |
| EMA1 |  |  | 0.886 |  |  |
| EMA2 |  |  | 0.867 |  |  |
| EMA3 |  |  | 0.859 |  |  |
| EMA4 |  |  | 0.900 |  |  |
| GI1 |  |  |  | 0.948 |  |
| GI2 |  |  |  | 0.948 |  |
| GI3 |  |  |  | 0.913 |  |
| GI4 |  |  |  | 0.916 |  |
| KM1 |  |  |  |  | 0.968 |
| KM2 |  |  |  |  | 0.988 |
| KM3 |  |  |  |  | 0.984 |
| KM4 |  |  |  |  | 0.984 |

**Table 3. Composite reliability.**

|  | Cronbach's alpha | Composite reliability | Average variance extracted (AVE) |
|---|---|---|---|
| BP | 0.910 | 0.951 | 0.782 |
| EC | 0.969 | 0.970 | 0.915 |
| EMA | 0.901 | 0.903 | 0.771 |
| GI | 0.949 | 0.949 | 0.868 |
| KM | 0.987 | 0.987 | 0.962 |

Fig 2 shows that the R² value for the estimated equation was 0.549, which is significant at the 1 percent level of probability. This means that 54.9% of the variation in bank performance is described by the EMA, KM, green innovation, and EC.

The next section shows the results of multicollinearity testing (Table 5 and Fig 3). As a rule of thumb, if the VIF values exceed 3.0, it indicates potential multicollinearity issues, which could affect the reliability of the regression estimates [75].

**Table 4. Discriminant validity.**

|  | BP | EC | EMA | GI | KM |
|---|---|---|---|---|---|
| BP | 0.884 |  |  |  |  |
| EC | 0.741 | 0.957 |  |  |  |
| EMA | 0.620 | 0.767 | 0.878 |  |  |
| GI | 0.524 | 0.626 | 0.75 | 0.932 |  |
| KM | 0.529 | 0.629 | 0.763 | 0.522 | 0.981 |

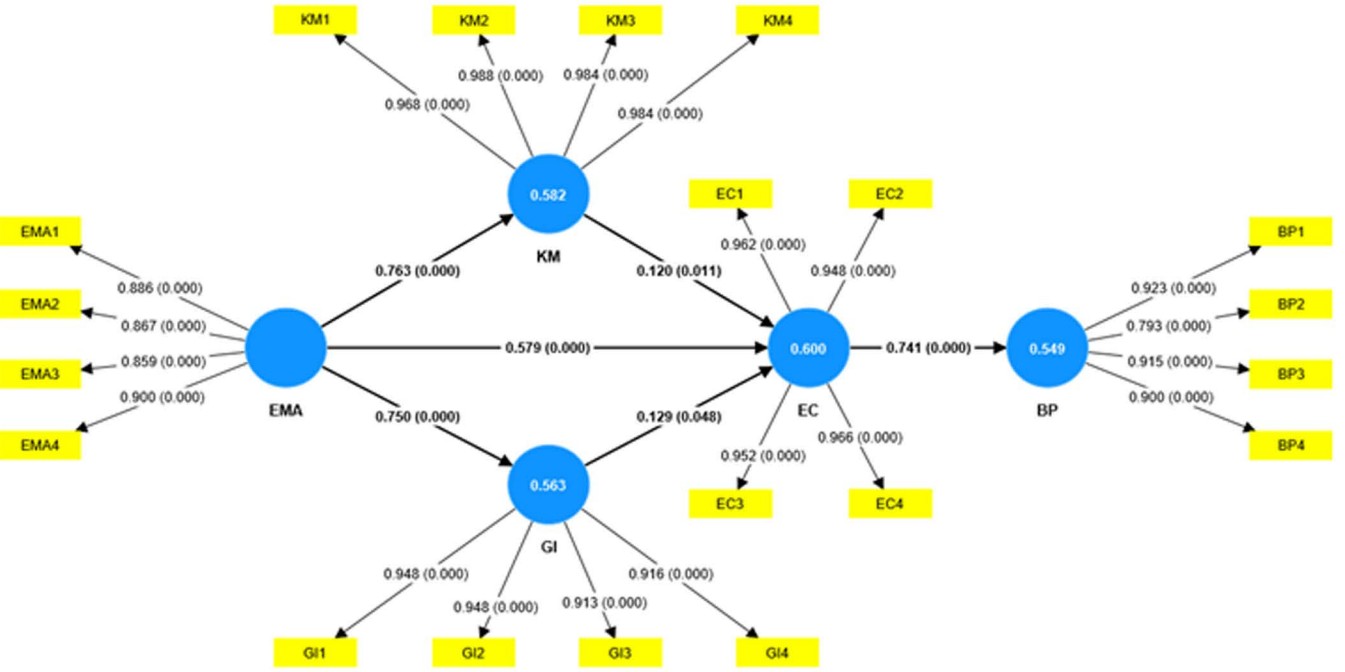

**Fig 2. Structural equation model (PLS-SEM).**

**Table 5. VIF values.**

|  | BP | EC | EMA | GI | KM |
|---|---|---|---|---|---|
| BP |  |  |  |  |  |
| EC | 1.000 |  |  |  |  |
| EMA |  | 2.043 |  | 1.000 | 1.000 |
| GI |  | 2.321 |  |  |  |
| KM |  | 2.427 |  |  |  |

All VIF values presented in Table 5 are below 3.0, indicating the absence of collinearity issues in the model. Additionally, collinearity can also be evaluated using the data presented in Fig 3. This figure provides a detailed breakdown of the VIF values for each construct.

This bar chart from Fig 3 presents the VIF values for each construct. The dashed red line at 3.0 indicates the typical threshold for multicollinearity concerns, showing that all VIF values are within acceptable limits.

## Discussions

Table 6 provides a comprehensive summary of the tested hypotheses, presenting the relationships between variables along with their corresponding beta coefficients, p-values, and evaluation results. The table clearly indicates the significance and strength of each hypothesized relationship. The findings in Table 6 show that the p-values of the supported hypotheses were statistically significant at the 5% level.

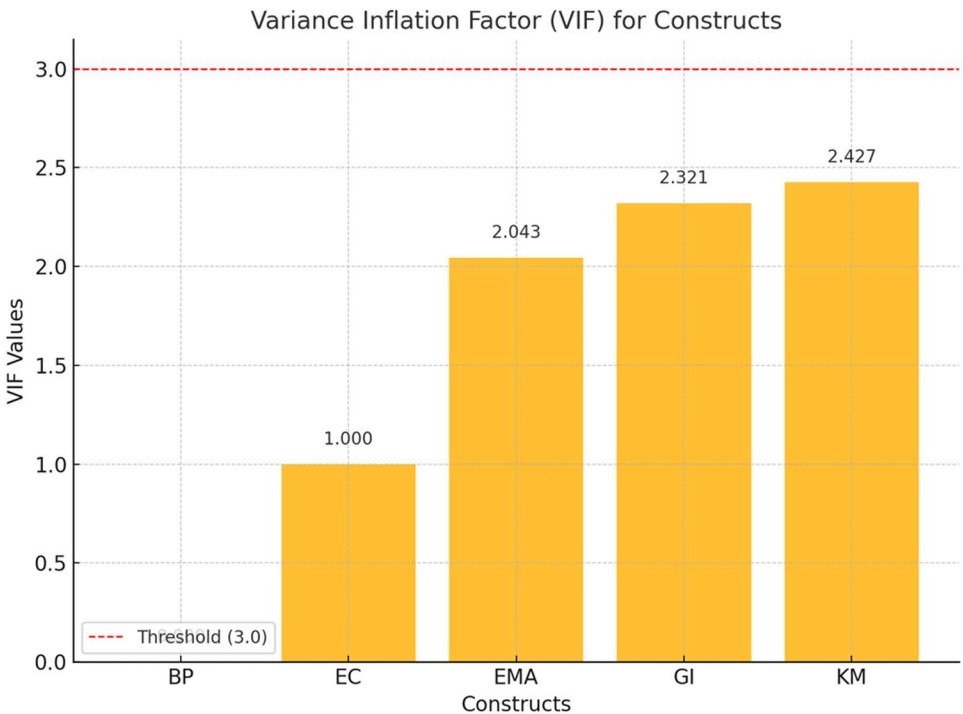

**Fig 3. VIF values for all constructs.**

**Table 6. Summary of hypotheses.**

| Path | Original sample (O) | Sample mean (M) | Standard deviation (STDEV) | T statistics (|O/STDEV|) | P values | Evaluate |
|------|---------------------|-----------------|----------------------------|--------------------------|----------|----------|
| EC -> BP | 0.741 | 0.742 | 0.031 | 24.171 | 0.000 | Accept |
| EMA -> EC | 0.579 | 0.574 | 0.073 | 7.916 | 0.000 | Accept |
| EMA -> GI | 0.750 | 0.748 | 0.046 | 16.202 | 0.000 | Accept |
| EMA -> KM | 0.763 | 0.761 | 0.052 | 14.620 | 0.000 | Accept |
| GI -> EC | 0.129 | 0.128 | 0.065 | 1.982 | 0.048 | Accept |
| KM -> EC | 0.120 | 0.126 | 0.047 | 2.530 | 0.011 | Accept |

As exhibited in Fig 4, EMA had a strong impact on KM (0.763), GI (0.750), and EC (0.579), indicating its critical role in the entire model. KM and GI also directly affect EC, but their coefficients are lower at 0.120 and 0.129, respectively, but they are significant contributing factors. Specifically, EC serves as both a dependent variable and a robust mediating variable, directly affecting BP with a path coefficient of 0.741, underscoring the fundamental construct of transforming environmentally friendly behavior into bank performance sustainability. The relationships indicate a clear directional hierarchy in which EMA drives KM, drives GI, enhances EC, and improves BP. Along with the summary statistics presented in Table 5, this figure provides an understanding of how the constructs interact to create a sustainable and competitive advantage.

Similarly, from Table 6, this study examines the role of EMA in improving bank performance through mediating factors such as knowledge management (KM), green innovation (GI), and environmental costs (EC). The findings reveal that EMA significantly enhances KM, GI, and EC, which, in turn, positively affect bank performance. These results provide robust evidence supporting the research hypotheses and addressing the core research questions by demonstrating the mechanisms through which EMA exerts its influence.

The study directly answers the research questions by elucidating the pathways linking EMA to bank performance. The results indicate a strong positive relationship between EMA and KM ($\beta = 0.763$, $p < 0.001$) as well as GI ($\beta = 0.750$, $p < 0.001$), underscoring EMA's ability to foster knowledge-sharing and innovative practices within banks. These findings highlight that EMA is not merely a financial tool but also a strategic enabler that drives organizational innovation and knowledge management capabilities. Furthermore, the strong impact of EC on bank performance ($\beta = 0.741$, $p < 0.001$) emphasizes the importance of environmental cost management as a key mediator, confirming its critical role in bridging

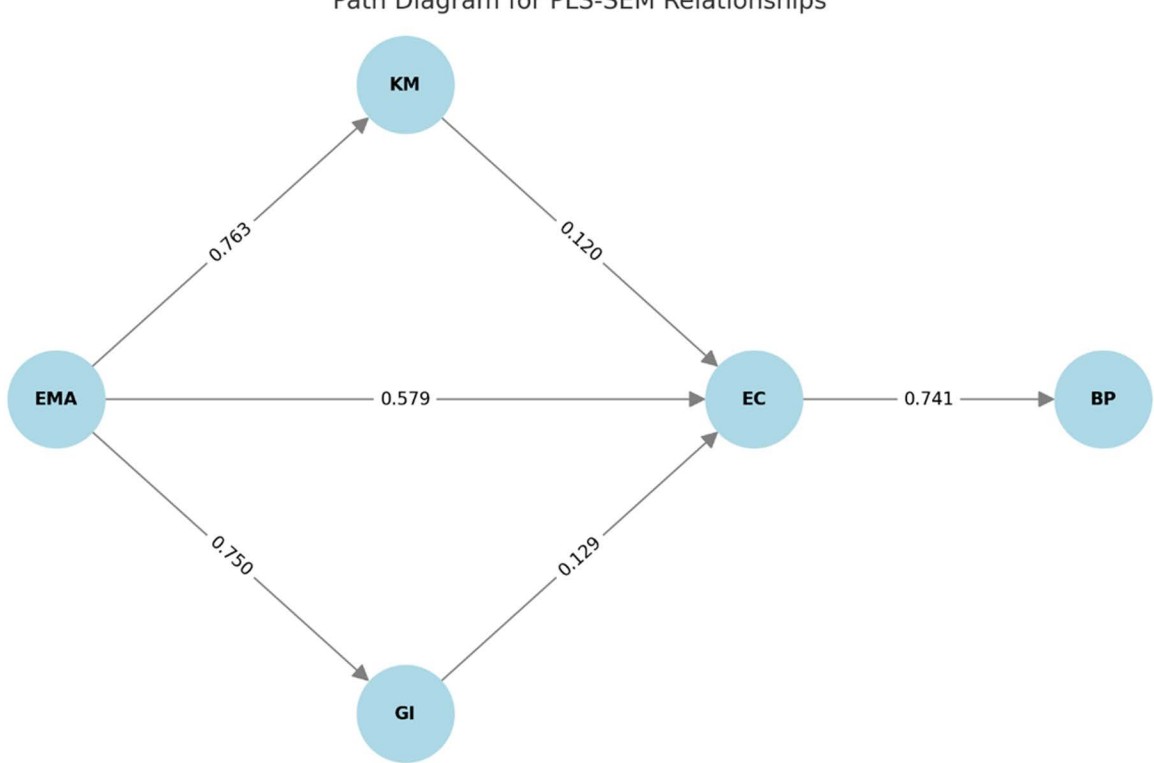

Path Diagram for PLS-SEM Relationships

**Fig 4. Path diagram for PLS-SEM relationships.**

sustainability initiatives with financial outcomes. By revealing these mechanisms, the study offers empirical validation for the conceptual model and enhances our understanding of how sustainability practices create value in the banking sector.

These findings are consistent with the literature while also providing new insights. EMA's observed effect on KM aligns with previous research by [8,22,23] within manufacturing contexts. However, this study explores the banking industry, where structural differences in financial operations exist. Likewise, the connection between EMA and GI confirms the findings reported by [29, 30], who highlighted that EMA complements green innovation. In contrast, by narrowing the scope to the Vietnamese banking sector, this research emphasizes the untapped potential of the banking sector as an environmentally pertinent source of credit allocation and resource mobilization, which are parameters frequently disregarded in previous literature. This small contribution highlights the importance of EMA as a mechanism to reconcile financial outcomes with sustainability targets, with a focus on emerging economies. Following that, this study examines whether the EC enhances the performance of commercial banks in Vietnam. The direct effect of EC on the performance of companies has been explored by [68] while the research examines whether EMA affects bank performance under the mediating role of EC.

This study makes important theoretical contributions by refining contingency theory and stakeholder theory, specifically within the context of EMA. The effectiveness of EMA is contingent upon industry-specific factors such as the technological and regulatory environment of Vietnamese banks, lending support to contingency theory. The results show the need to integrate EMA schemas into the local context to harness the most benefit. Stakeholder theory is again supported by demonstrating how transparency and environmental accountability–underlined by strong EMA–lead stakeholders to have more stake in the company, which, in turn, feeds into the financial performance of the company. Advancing theoretical perspectives on the realization of EMA in various organizations and ecological modalities in this manner leads to a broad-based platform for the applications of EMA.

## Conclusions and limitations

This study assesses the impact of EMA on bank performance, focusing on how this influence is mediated by EC. The author conducted the research using qualitative and quantitative methods. The findings show that the EMA positively affects KM, green innovation, and environmental costs in the Vietnamese banking sector. In addition, the EC is influenced by KM and green innovation by 5%. In particular, EC is a statistically significant predictor of performance in commercial banks in Vietnam. Embedded within environmental accounting, EMA functions as a valuable instrument to overcome the constraints inherent in traditional management accounting. This fosters a more profound comprehension of social responsibility and aids in shaping environmentally conscious practices. EMA not only enhances a company's societal standing but also plays a pivotal role in effectively managing environmental responsibilities by efficiently controlling the associated EC. Consequently, this approach can lead to substantial improvement in bank performance.

Based on the findings, we propose some suggestions for applying EMA in Vietnamese commercial banks. First, banks ought to integrate EMA frameworks in their strategic governance systems by creating in-house systems that capture and integrate environmental cost data. Such tools consist of the use of advanced technologies (for example: data analytics, artificial intelligence) to help improve the reliability of environmental cost tracking and reporting. Second, the managerial know-how on environmental accounting should be augmented through training programs so that human resources, as a labour force becomes proficient in this innovative management technique. Third, the banks can fund green innovation projects, including energy-efficient technologies and sustainable product development, to emphasize their support of environmental sustainability.

Some findings are specific to this study, which should be aware of some limitations. First, the small sample size contributes to one limitation, thus the need for future research that can use a larger, more heterogeneous sample. Second, more research should be conducted exploring the long-term influence of EMA on financial stability and its broader consequences for society in developing countries like Vietnam. The research may also explore how digital transformation

enables EMA adoption and the impact on new sustainability practices. Furthermore, investigating the role of regulatory and technological factors in adopting EMA among several banking systems would yield significant insight for adopting suitable practices. Finally, more mediators, such as corporate social responsibility and environmental risk management, may clarify the relationship between EMA and bank performance.

## Appendix

### Questionnaire

| No. | Factors/ Observed variables | Code | Source |
|---|---|---|---|
| **I** | **Environmental management accounting** | **EMA** | [4,77] |
| 1 | Environmental management accounting provides information for strategic decision-making. | EMA1 | |
| 2 | Banks are increasingly interested in environmental management accounting. | EMA2 | |
| 3 | Environmental management accounting helps banks expand market share. | EMA3 | |
| 4 | Environmental management accounting helps enhance the image and position of the bank. | EMA4 | |
| **II** | **Knowledge management** | **KM** | [78,79] |
| 1 | Managers in various units prioritize benefits, thus they have a heightened interest in environmental cost information as well as environmental cost management. | KM1 | |
| 2 | Developing a unified documentation system to capture knowledge on environmental management accounting. | KM2 | |
| 3 | The capability to manage knowledge is transferred into the business efficiency of the organization. | KM3 | |
| 4 | The bank regularly undertakes efforts to enhance the effectiveness of knowledge management. | KM4 | |
| **IV** | **Green innovation** | **GI** | [53] |
| 1 | The bank consistently addresses energy reduction, resource consumption, and waste minimization throughout the process of delivering financial products and services to customers. | GI1 | |
| 2 | The bank continuously promotes the development and adoption of green solutions, technologies, and products, meeting various standards. | GI2 | |
| 3 | The bank consistently prioritizes funding and support for green projects, sponsors energy-saving initiatives, and develops projects to combat climate change. | GI3 | |
| 4 | The bank consistently identifies its role and responsibility in "greening" investment capital for sustainable development objectives. | GI4 | |
| **V** | **Environmental costs** | **EC** | [80] |
| 1 | The bank complies with environmental laws by balancing environmental costs and benefits achieved. | EC1 | |
| 2 | Environmental costs often represent a relatively significant proportion of the overall operational costs of the bank's business activities. | EC2 | |
| 3 | Environmental cost information is often not fully disclosed and reported. | EC3 | |
| 4 | Environmental costs are an integral part of environmental management accounting. | EC4 | |
| **VI** | **Bank performance** | **BP** | [53,77] |
| 1 | Implementing environmental management accounting helps the bank maintain business efficiency. | BP1 | |
| 2 | The bank consistently fulfills its environmental responsibility through standards and changes in the accounting system. | BP2 | |
| 3 | Compliance with environmental laws enables the bank to operate legally and effectively in its business activities. | BP3 | |
| 4 | Enhancing awareness, role, and capacity of the banking sector in providing credit to green sectors/fields, managing environmental and social risks in credit activities. | BP4 | |

# Author contributions

**Conceptualization:** Kim Quoc Trung Nguyen.

**Data curation:** Kim Quoc Trung Nguyen.

**Formal analysis:** Kim Quoc Trung Nguyen.

**Funding acquisition:** Kim Quoc Trung Nguyen.

**Investigation:** Kim Quoc Trung Nguyen.

**Methodology:** Kim Quoc Trung Nguyen.

**Project administration:** Kim Quoc Trung Nguyen.

**Resources:** Kim Quoc Trung Nguyen.

**Software:** Kim Quoc Trung Nguyen.

**Supervision:** Kim Quoc Trung Nguyen.

**Validation:** Kim Quoc Trung Nguyen.

**Visualization:** Kim Quoc Trung Nguyen.

**Writing – original draft:** Kim Quoc Trung Nguyen.

**Writing – review & editing:** Kim Quoc Trung Nguyen.

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
