## [Decision Letter · Decision Letter 0]

1 Dec 2024

Dear Dr. Nguyen,

Thank you for submitting your manuscript to PLOS ONE. After careful consideration, we feel that it has merit but does not fully meet PLOS ONE’s publication criteria as it currently stands. Therefore, we invite you to submit a revised version of the manuscript that addresses the points raised during the review process.

We look forward to receiving your revised manuscript.

Kind regards,

Reza Rostamzadeh

Academic Editor

PLOS ONE

Journal Requirements:

2. We note that your Data Availability Statement is currently as follows: [All relevant data are within the manuscript and its Supporting Information files.] Please confirm at this time whether or not your submission contains all raw data required to replicate the results of your study. Authors must share the “minimal data set” for their submission. PLOS defines the minimal data set to consist of the data required to replicate all study findings reported in the article, as well as related metadata and methods (https://journals.plos.org/plosone/s/data-availability#loc-minimal-data-set-definition). For example, authors should submit the following data: - The values behind the means, standard deviations and other measures reported; - The values used to build graphs; - The points extracted from images for analysis. Authors do not need to submit their entire data set if only a portion of the data was used in the reported study. If your submission does not contain these data, please either upload them as Supporting Information files or deposit them to a stable, public repository and provide us with the relevant URLs, DOIs, or accession numbers. For a list of recommended repositories, please see https://journals.plos.org/plosone/s/recommended-repositories. If there are ethical or legal restrictions on sharing a de-identified data set, please explain them in detail (e.g., data contain potentially sensitive information, data are owned by a third-party organization, etc.) and who has imposed them (e.g., an ethics committee). Please also provide contact information for a data access committee, ethics committee, or other institutional body to which data requests may be sent. If data are owned by a third party, please indicate how others may request data access.

Reviewers' comments:

Reviewer's Responses to Questions

**Comments to the Author**

1. Is the manuscript technically sound, and do the data support the conclusions?

Reviewer #1: Yes

Reviewer #2: No

Reviewer #3: Yes

2. Has the statistical analysis been performed appropriately and rigorously?

Reviewer #1: Yes

Reviewer #2: No

Reviewer #3: Yes

3. Have the authors made all data underlying the findings in their manuscript fully available?

Reviewer #1: Yes

Reviewer #2: No

Reviewer #3: Yes

4. Is the manuscript presented in an intelligible fashion and written in standard English?

Reviewer #1: Yes

Reviewer #2: Yes

Reviewer #3: Yes

Reviewer #1: The paper investigates the influence of Environmental Management Accounting (EMA) on the performance of Vietnamese commercial banks, focusing on the mediating role of environmental costs (EC), knowledge management (KM), and green innovation (GI). Utilizing a mixed-methods approach, the study employs qualitative expert interviews and quantitative Partial Least Squares Structural Equation Modeling (PLS-SEM) to analyze data from 312 respondents. The findings suggest significant positive relationships between EMA and KM, GI, and EC, as well as between these mediators and bank performance. The paper emphasizes the importance of integrating EMA into sustainable banking practices to enhance performance and environmental responsibility.

Strengths:

1. Relevance and Contribution: The study addresses a critical and underexplored topic, particularly in the context of Vietnam, where EMA practices are relatively new.

2. Methodological Rigor: The use of a mixed-methods approach and robust statistical tools (PLS-SEM) ensures comprehensive and credible findings.

3. Practical Implications: The paper offers actionable insights into how banks can integrate EMA and sustainability practices to improve performance.

Weaknesses and Recommendations:

1. Introduction and Contextualization:

o The introduction provides a general overview but lacks a compelling narrative to highlight the research gap.

o Recommendation: Clearly state the novelty of the study and how it addresses specific gaps in EMA research within the Vietnamese banking sector.

2. Literature Review:

o The literature review is extensive but occasionally redundant, especially in repeating the definitions of EMA and EC.

o Recommendation: Streamline the review to focus on critical theories and studies directly relevant to the hypotheses.

3. Research Methodology:

o The sample size of 312 is adequate but might benefit from more diversity in bank types or geographical representation.

o Recommendation: Discuss the potential impact of sample limitations on generalizability.

4. Data Analysis and Presentation:

o While the statistical analysis is thorough, the presentation of results (e.g., VIF values, discriminator validity) can be made more accessible with clearer visual aids.

o Recommendation: Include more graphical representations, such as path diagrams for the PLS-SEM model.

5. Conclusion and Practical Implications:

o The conclusion reiterates the findings but lacks a forward-looking perspective on future research and practical applications.

o Recommendation: Provide specific strategies for banks to implement EMA practices and suggest areas for further research.

6. Language and Formatting:

o There are minor grammatical errors and inconsistencies in formatting.

o Recommendation: Conduct a thorough language review to ensure clarity and adherence to journal guidelines.

Reviewer #2: I appreciate the opportunity that was given to me by the editor to review the article "Environmental Management Accounting Affects Bank Performance with Mediators". I regret to say that after thoroughly reviewing the manuscript, I recommend its rejection based on the following critical issues:

Introduction

• The introduction lacks an adequate background that contextualizes the study. Key concepts related to environmental accounting and its relevance to the environmental impacts being examined are not discussed, leaving readers without a clear understanding of the study's foundation.

• A well-defined research problem is essential to frame the study, yet the manuscript does not articulate any specific problem. Without a clear problem statement, the motivation and necessity of the research are not evident.

• The manuscript does not include a meaningful review of prior studies. There is no critical analysis of existing research, no identification of gaps, and no explanation of how this study differs from or builds upon previous work. This oversight diminishes the manuscript’s scholarly contribution.

• The introduction does not articulate the theoretical or practical contributions of the research. It is unclear how this study adds value to the academic field or to practical applications in environmental accounting.

• The research is conducted in the context of Vietnamese banks, yet there is no rationale for this choice. The study does not explain how banks contribute to or mitigate environmental impacts, which creates a significant disconnect between the topic and the case selection.

• The introduction relies on a limited number of references, many of which are outdated. The use of older references weakens the validity of the claims made and does not reflect the current state of research in the field. To substantiate the research arguments, the manuscript needs to draw from a broader and more recent body of literature.

Literature Review

The section on prior studies is overly detailed, the extensive focus on the literature review appears to have come at the expense of other essential sections, such as methodology and discussion. These sections lack sufficient detail and depth, which undermines the overall quality of the manuscript.

Methodology

Lack of Clarity in the Mixed-Method Approach:

• While the authors claim to have adopted a mixed-methods approach that integrates qualitative and quantitative research techniques, the steps for conducting the qualitative component are not clearly outlined. The absence of detailed procedures, such as data collection, sampling, or coding methods, creates ambiguity and casts doubt on the reliability of the qualitative findings.

• According to Hair et al., (2017), Structural Equation Modeling (SEM) consists of two main components:

• Measurement Model: This part describes the relationship between latent variables (constructs that are not directly observable) and their corresponding observed indicators. It allows researchers to assess the validity and reliability of the constructs.

• Structural Model: This part illustrates the causal relationships between the latent variables, showing how they influence one another based on theoretical assumptions.

• The authors have ignored the important part of the measurement of the model which is Factor loadings. Factor loadings of ≥ 0.70 contribute to convergent validity, demonstrating that the indicators of a construct are highly related to one another. This ensures that the construct is well-defined by its indicators. The ignoring of this part leads to invalid results in the structural model.

• In addition, the structural model lacks a lot of information that must be presented to ensure the validity of the results, such as standard deviation, Confidence Interval, and t-value.

• In general, the methodology is unclear and full of ambiguity, which requires a complete reconsideration.

Issues in the Discussion Section

• The discussion is written descriptively without a clear link between the results and the research questions. The findings are presented, but there is little effort to explain how they address the core questions of the study or how they contribute to understanding the research problem. This disconnect weakens the paper's coherence and its ability to advance knowledge in the field.

• The discussion section does not adequately compare or contrast the findings with existing research. There is no reference to how the results align with or differ from previous studies, which is essential for contextualizing the contribution of the current research. A thorough comparison with past studies would highlight the novel aspects of this research and its relevance to the field.

• The discussion does not explicitly address the theoretical contributions of the research. The study’s implications for theory are not discussed, nor is there an explanation of how the findings contribute to the development or refinement of the theoretical framework used in the study. Without this discussion, the theoretical significance of the research remains unclear.

• The practical implications of the research are not addressed in the discussion. Given the applied nature of the study, it is crucial to outline how the findings can inform practice in the field, such as in environmental accounting or bank management. The lack of practical insights limits the relevance of the study to practitioners and policymakers.

These issues significantly weaken the manuscript’s contribution to both theory and practice. The discussion needs to be restructured to clearly link the findings to the research questions, engage with the relevant literature, and outline both theoretical and practical contributions.

Reviewer #3: the manuscript " Environmental Management Accounting Affects Bank Performance with Mediators" is sound

My decision is to accept the manuscript

The writing is wonderful and expressive

The analysis that has been done is appropriate

There is no objection to accepting it for publication in the magazine

From my point of view

**Do you want your identity to be public for this peer review?** For information about this choice, including consent withdrawal, please see our Privacy Policy

Reviewer #1: **Yes: ** Dr.Omar Alsinglawi

Reviewer #2: No

Reviewer #3: No

---

## [Author Response · Author response to Decision Letter 1]

8 Feb 2025

Dear Sir/Madam,

I have updated the manuscript in accordance with the reviewers' comments. The revised manuscript and the Response to Reviewers document are attached for your reference.

Thank you and best regards,

Kim Quoc Trung Nguyen

---

## [Decision Letter · Decision Letter 1]

3 Mar 2025

Environmental Management Accounting Affects Bank Performance with Mediators

PONE-D-24-30039R1

Dear Dr. Nguyen,

We’re pleased to inform you that your manuscript has been judged scientifically suitable for publication and will be formally accepted for publication once it meets all outstanding technical requirements.

Kind regards,

Reza Rostamzadeh

Academic Editor

PLOS ONE

Additional Editor Comments (optional):

Reviewers' comments:

Reviewer's Responses to Questions

**Comments to the Author**

Reviewer #1: All comments have been addressed

Reviewer #2: All comments have been addressed

2. Is the manuscript technically sound, and do the data support the conclusions?

Reviewer #1: Yes

Reviewer #2: Yes

3. Has the statistical analysis been performed appropriately and rigorously?

Reviewer #1: Yes

Reviewer #2: Yes

4. Have the authors made all data underlying the findings in their manuscript fully available?

Reviewer #1: (No Response)

Reviewer #2: Yes

5. Is the manuscript presented in an intelligible fashion and written in standard English?

Reviewer #1: Yes

Reviewer #2: Yes

Reviewer #1: Your study provides critical insights into EMA and its impact on bank performance. The findings are significant, and the methodology is well-executed.

Reviewer #2: (No Response)

**Do you want your identity to be public for this peer review?** For information about this choice, including consent withdrawal, please see our Privacy Policy

Reviewer #1: **Yes: ** Dr.Omar Alsinglawi

Reviewer #2: No

---

## [Editor Report · Acceptance letter]

PONE-D-24-30039R1

PLOS ONE

Dear Dr. Nguyen,

I'm pleased to inform you that your manuscript has been deemed suitable for publication in PLOS ONE. Congratulations! Your manuscript is now being handed over to our production team.

Kind regards,

on behalf of

Dr. Reza Rostamzadeh

Academic Editor

PLOS ONE